# Ceramide and Sphingosine Regulation of Myelinogenesis: Targeting Serine Palmitoyltransferase Using microRNA in Multiple Sclerosis

**DOI:** 10.3390/ijms20205031

**Published:** 2019-10-11

**Authors:** Somsankar Dasgupta, Swapan K. Ray

**Affiliations:** 1Department of Neuroscience and Regenerative Medicine, Augusta University, 1120 15th Street, Augusta, GA 30912, USA; 2Department of Pathology, Microbiology, and Immunology, University of South Carolina School of Medicine, 6439 Garners Ferry Road, Columbia, SC 29209, USA; Swapan.Ray@uscmed.sc.edu

**Keywords:** ceramide, development, microRNA, multiple sclerosis, MS therapy, oligodendrocytes protection, serine palmitoyltransferase, sphingosine

## Abstract

Ceramide and sphingosine display a unique profile during brain development, indicating their critical role in myelinogenesis. Employing advanced technology such as gas chromatography–mass spectrometry, high performance liquid chromatography, and immunocytochemistry, along with cell culture and molecular biology, we have found an accumulation of sphingosine in brain tissues of patients with multiple sclerosis (MS) and in the spinal cord of rats induced with experimental autoimmune encephalomyelitis. The elevated sphingosine leads to oligodendrocyte death and fosters demyelination. Ceramide elevation by serine palmitoyltransferse (SPT) activation was the primary source of the sphingosine elevation as myriocin, an inhibitor of SPT, prevented sphingosine elevation and protected oligodendrocytes. Supporting this view, fingolimod, a drug used for MS therapy, reduced ceramide generation, thus offering partial protection to oligodendrocytes. Sphingolipid synthesis and degradation in normal development is regulated by a series of microRNAs (miRNAs), and hence, accumulation of sphingosine in MS may be prevented by employing miRNA technology. This review will discuss the current knowledge of ceramide and sphingosine metabolism (synthesis and breakdown), and how their biosynthesis can be regulated by miRNA, which can be used as a therapeutic approach for MS.

## 1. Introduction

Sphingolipids are amphipathic molecules with diverse functions in cell development and death. Ceramide (Cer) and sphingosine (Sph) are two unique sphingolipids that comprise the core structural bases for all sphingolipids. The biological functions of sphingoids such as Sph, dihydrosphingosine (dhSph), and psychosine as well as of sphingolipids are regulated by their relative concentrations in cells and tissues. For example, Cer participates in a wide variety of biochemical events such as protein phosphorylation, modulation of protein kinase C (PKC) and phospholipase A_2_, but mediates signal transduction leading to cell death at a high concentration, while Sph and psychosine are potent inhibitors of PKC [1]. Hence, the precise concentration of sphingolipid metabolites is the key to normal tissue development and maintenance. Besides their important regulatory functions in brain development and myelinogenesis [1], they also actively contribute to cell death, including promotion of cell death in oligodendrocytes in multiple sclerosis (MS) [2].

MS is a devastating demyelinating human disease leading to extensive neurodegeneration and is characterized by immune mediated loss of myelin and axons [3]. The clinical manifestations and disease progression in MS are substantially variable and related to complex genetic, epigenetic, and environmental factors [4,5,6,7]. MS is generally characterized by the infiltration of lymphocytes into the central nervous system (CNS), leading to axonal demyelination and degeneration of neurons [8]. The development of modern genomics has successfully identified the various molecular mechanisms in MS pathogenesis [9]. The down- and up-regulation of specific microRNAs (miRNAs) may nowadays be used as biomarkers for predicting disease progression [10]. Identifying agents inhibiting the enzymes controlling epigenetic modifications, particularly DNA methyltransferases and histone deacetylases, may be introduced as promising tools for therapeutic interventions in MS [11].

Although multiple factors may have been implicated in the pathogenesis of MS, all reports indicate a severe change in the biochemical milieu in the CNS in progression of MS. The inflammatory microenvironment contains a variety of substances such as proteolytic enzymes, cytokines, oxidative products, and free radicals that can injure neurons and axons [12]. Studies including human and animal models of MS have indicated that inflammatory cytokines play a critical role in MS pathogenesis [13] by fostering inflammation and demyelination [10]. Cytokines such as tumor necrosis factor alpha (TNFα) and interferon gamma (IFNγ) mediate aberrant lipid metabolism with accumulation of toxic Sph in MS brains [2,14]. This, in turn, may further damage brain function due to degeneration of oligodendrocytes and neurons, the two key CNS cell types for myelin constituents [2]. Investigations have shown that an intermittent increase in Cer by TNFα and IFNγ stimulation of serine palmitoyltransferase (SPT) activity followed by Sph accumulation in spinal cords from Lewis rats with experimental autoimmune encephalitis (EAE) resulted in induction of apoptosis in the lumbar spinal cord [2]. TNFα and IFNγ stimulated Cer elevation in cultured human oligodendrocytes and Cer production was blocked by myriocin, an inhibitor of SPT, causing prevention of apoptosis in oligodendrocytes [2].

Medications are used in MS to modify the disease course and manage MS symptoms. Although there is no cure, the medications are apparently essential components that help people manage their MS and enhance their life. Such medications include oral administration of teriflunomide, fingolimod, dimethyl fumarate, injectable administrations (interferon beta 1a, glatirmer acetate, and daclizumab), and infused medications (alemtuzumab, mitoxantrone, and natalizumab) [15]. Symptomatic medications are also available to treat syndromes such as bladder problems, fatigue, emotional stress, etc. Although many clinical trials are being conducted for MS, to the best of our knowledge, there is no primarily sphingolipid-targeted therapy available, with the exception of fingolimod, which acts as a ligand for the Sph-1-P receptor and inhibits Cer synthase [2]. Hence, Sph toxicity via Cer generation may be a potential target in both EAE and MS as an alternate therapy by blocking Cer synthesis.

In this article, we introduce the concept of miRNA therapy to protect the oligodendrocytes from apoptosis in order to prevent demyelination in MS. Expression of several miRNAs such as miR-155 and miR-326 are increased in MS brains and these may have potential roles in MS pathogenesis [3]. Several factors that contribute to MS pathogenesis foster aberrant lipid metabolism leading to accumulation of Sph, a toxic sphingolipid, that has been demonstrated to mediate apoptosis in human gastric cancer cells [16], hepatoma cells [17], and rhabdomyosarcoma cells [18]. Our recent study demonstrated an up-regulation of Sph in MS tissues causing demyelination via oligodendrocyte apoptosis [2].

There are many therapeutic strategies available for MS based on the patient’s condition. Although fingolimod is a lipid inhibitor, its efficacy in inhibiting Cer generation is much less than that of myriocin [2]. Myriocin itself displays toxicity in humans and hence is unsuitable for human therapy. Other natural or synthetic inhibitors of Cer synthesis such as fumonisin B1 may display severe adverse effects in humans [19]. Hence, an alternate therapy that involves the use of miRNA may show high potential. At present multiple effects of miRNAs are well known, and they are considered to be nontoxic (or less toxic) because of their natural synthesis in humans. This review describes the basic rationale of miRNA therapy to regulate Cer/Sph accumulation in MS as a novel therapeutic approach for MS.

## 2. Diversity in Sphingolipid Functions

### 2.1. Ceramide (Cer) and Sphingosine (Sph) Stimulate Myelinogenesis

To determine the regulation of sphingolipids in myelinogenesis, we quantified the concentrations of Cer, Sph/dhSph, and monoglycosylceramides (MGCs) during rat brain development using our advanced methodology [20] and showed that all three components are critical for myelination. An increasing concentration of Cer during development, with an optimum concentration at postnatal day (P) 21, clearly signifies its direct involvement during myelination. However, its precise mechanism (i.e., interactions with other myelin components) has not yet been explored. Cer, being the core structure, may stimulate the synthesis of other myelin-specific sphingolipids such as GalCer, GM1, etc. that are pertinent to the interactive composition of myelin. Interestingly, the Cer/dihydro-(dh) Cer ratio determined using gas chromatography–mass spectrometry (GC–MS) and high performance liquid chromatography (HPLC) indicated a constant value between 4.0/1 and 4.5/1 during normal brain development. A higher ratio has been observed in MS brains where an aberrant sphingolipid metabolic activity has recently been confirmed [2].

For MGC, GlcCer is the only component in embryonic and early postnatal brains (P2 to P5). GalCer synthesis initiates during subsequent developmental stages (P8–P10) with increasing concentration, reaches an optimum at P21, and then stabilizes. The activities of two synthesis enzymes, Cer:glucosyltransferase and Cer:galactosyltransferase were in correspondence with the GlcCer and GalCer concentrations [21]. The GalCer concentration in MS brain tissue has been greatly reduced as a direct manifestation of extensive demyelination [22]. A schematic presentation of sphingolipid-mediated myelinogenesis is shown (Figure 1).

Similarly, Sph/dhSph concentrations are gradually elevated until P10 during the stage of (oligodendrocyte) development and then they decrease. A secondary short peak is seen during myelinogenesis at P21 with a steady ratio (2.2–2.5/1) of Sph/dhSph, which varies greatly in MS brains [1]. Cer is produced from dhCer (by desaturase or DES), which in turn is hydrolyzed to Sph by ceramidase (Figure 2). The accumulation of Sph may be triggered by excess Cer biosynthesis (as a breakdown product of ceramidase) as shown in the diagram (Figure 2). Sph plays an unknown but critical role in oligodendrocyte synthesis and maturation during myelination by activating the melastatin-like transient receptor potential protein 3 (TRMP3) [23] which facilitates spontaneous Ca^2+^ entry. Hence, the appearance of the peak at P10 and then a short peak at P21 [1] correlates with the activation of TRMP3, as TRPM3 participates as a Ca^2+^-permeable and Sph-activated channel in oligodendrocyte differentiation (P10) and CNS myelination [23]. This corresponds with our observation that a lower Sph concentration stimulates human oligodendrocyte proliferation, while a higher concentration leads to cell death [2,24]. However, the detailed mechanism by which Cer and Sph aid myelinogenesis needs further investigation.

We have recently described the chromatographic purification and characterization of phytoCer in vertebrate brains and other tissues [25], indicating that a more careful evaluation of Cer biosynthesis is warranted as no such naturally-occurring Cer species has been reported in vertebrates. We postulate that phytoCer may be biosynthesized by addition of a fatty acyl group to phytoSph by a phytoCer synthase, and this enzyme has not yet been reported [1]. Moreover, a huge accumulation of Sph at an initial developmental stage (P10) without any major peaks for Cer leads us to predict that a DES may possibly exist that directly converts dhSph into Sph [1].

### 2.2. Sphingolipid Metabolic Disorders

Sphingolipid metabolic diseases are designated as inborn errors of metabolism, which include Gauscher’s and Krabbe’s disease [26,27,28], GM1 gangliosidosis [29], Tay-sach’s disease [30], etc. In these diseases, there is an accumulation of a specific sphingolipid in a particular tissue(s)/organ(s) due to the deficiency of a specific metabolic enzyme. However, no such disease correlating Cer or Sph accumulation has yet been reported. Cer has been at the center of extensive study for its role in various cell death mechanisms in nervous system disorders [31,32,33,34]. We have recently reported Sph elevation preceeded by a transient accumulation of Cer in MS brains [2]. This mechanism is due to cytokine stimulated SPT activation, which is an initial step of the pathway leading to Cer biosynthesis (Figure 2), rather than sphingomyelin degradation (salvage pathway). Although Sph has been shown to trigger oligodendrocyte cell death, the precise mechanism remains uninvestigated.

### 2.3. Aberrant Lipid Metabolism in MS Brains

Demyelination in MS may proceed via different mechanisms in different MS patients due to its stimulation by a variety of factors such as proteolytic enzymes, cytokines, oxidative products, and free radicals [18]. It is evident that inflammatory cytokines play a critical role in pathogenesis in MS lesions in humans and animals [35,36] by triggering inflammation [12,36], although the mechanism of cytokine action remains unknown. Immune cell filtration accompanying chronic inflammation signifies that MS is an autoimmune demyelinating disease, even though the precise etiology of MS has not yet been resolved [37]. Moreover, many major issues such as the primary cause of inflammation, primary target antigen, stimulation of autoimmunity, etc. have not yet been answered [38,39,40].

The CNS, more specifically, myelin, is enriched with sphingolipids and lysosphingolipids such as GalCer and Cer, sphingoids, and psychosine [20]. An increasing concentration of these sphingolipids during brain development and myelinogenesis implies that (lyso)sphingolipids may participate in cell growth, differentiation, myelinogenesis, and maintenance of the structural and functional integrity of myelin [1,20]. However, their concentration in white matter of a normal brain, in normal appearing white matter (NAWM) and in a plaque of a MS brain significantly varies [2]. An accumulation of psychosine and increasing trend of Sph has been reported in NAWM, while MS plaque shows a decrease in Cer concentration but an accumulation of Sph. The breakdown of Cer by ceramidase is the prime source of Sph production, while cytokine-mediated activation of SPT plays the major role in Cer synthesis in vivo (Figure 2). Although there are few studies involving the sphingolipid/lipid metabolism in MS [41,42,43], our recent study [2] has shown the detailed changes in sphingolipid profile with a dramatic accumulation of Sph in patient brains and EAE animal spinal cords, and this is in agreement with a recent publication [44].

Briefly, we have shown that in addition to Galcer, both Cer and Sph are necessary components for myelinogenesis, maintaining a defined ratio of Cer/dhCer and Sph/dhSph during the rat brain development and a higher ratio is observed in MS due to accumulation of Cer and Sph that lead to oligodendrocyte death and thereby fostering demyelination.

### 2.4. Expression of miRNAs

The small non-coding RNAs, which have an average length of 22 nucleotides, are known as miRNAs. All miRNAs are evolutionarily conserved and have been detected only in eukaryotic cells. The first miRNA was discovered in *Caenorhabditis elegans*. It showed complementarity to the 3’ untranslated region (3’UTR) of a target messenger RNA (mRNA) for a RNA–RNA interaction and its degradation (silencing) [45]. This discovery triggered many other studies, which concluded that all miRNAs were single stranded RNA molecules (ssRNA). They have a stem-loop structure. All miRNAs are highly stable and are partially complementary to the 3’UTR of the target mRNAs. Moreover, all miRNAs can act on multiple target mRNAs. Studies have shown that each miRNA may degrade almost 100 different target mRNAs. All miRNAs form a network of complex regulation. A specific mRNA may contain multiple binding sites for many miRNAs. Intergenic regions and introns of protein-coding genes most frequently contain miRNA genes. It now appears that so called ‘junk DNA’ locations act as the origin of most of the miRNAs. Studies have revealed that biogenesis and maturation of miRNAs occur in four steps: transcription, nuclear processing, nuclear exportation, and cytoplasmic processing [45]. It is now well-established that all miRNAs are highly conserved molecules and play highly regulatory roles in activation or inhibition of many cell-signalling pathways during normal development. Deregulation of miRNAs causes diseases.

#### 2.4.1. miRNAs in Oligodendrocyte Development and Ceramide Metabolism

Although significant progress has been made in examining lipid metabolism, little effort is made to investigate the role of miRNAs in sphingolipid metabolism [46]. Oligodendrocyte (oligo) progenitors express A2B5 antigen, while GalC is the considered as the matured oligo marker. The miRNA expression profile in oligo lineage cells is known as “microRNAome” that contains 43 miRNAs [47] and their expression changes from oligo progenitor cells (A2B5) to pre-myelinating oligo (GalC^+^) cells that include a target bias for a class of miRNA (including miR-9). MiR-9 expression is down-regulated during differentiation of oligos and its expression inversely correlates with the expression of its target which is the peripheral myelin protein 22 (PMP22) [47]. By interacting with the PMP22, miR-9 down-regulates the expression of PMP22 [47]. SPT is the first limiting enzyme of de novo Cer synthesis (Figure 2) and its precise regulation is still obscure. A recent study examining the regulation of SPT in Alzheimer’s disease identified the loss of miR-137, miR-181c, miR-9, and miR-29a/b-1, which stimulated SPT and Aβ levels [48]. Moreover, FTY720 increases miR-376, miR-30, miR-128, miR-126, miR-7, and miR-9, interfering with Cer synthesis [49]. Hence, SPT down-regulation by miRNA technology may be considered as a novel approach to protect the oligos from Cer/Sph toxicity.

#### 2.4.2. miRNAs in MS and Neurodegenerative Disorders

Current studies strongly indicate alterations in expression of neurodegenerative and neuroprotective miRNAs in neurodegenerative injuries or diseases including MS [50,51,52]. Levels of neurodegenerative miRNAs are increased, while levels of neuroprotective miRNAs are down-regulated, contributing to progressive neurodegeneration. Fumonisin B_1_, a naturally occurring myotoxin in maize and an inhibitor of Cer synthesis significantly down-regulated expression of miR-27 in HepG2 cells, while human cytochrome P450 mRNA and protein expression were up-regulated in HepG2 cells [53]. Therefore, levels of specific miRNAs can serve as important biomarkers for diagnosis, treatment, and prognosis in a CNS injury or disease [54]. Recently, a radical miRNA-based therapy has been designed to encompass neuroprotection and neuronal restoration [55,56].

#### 2.4.3. Role of miRNA in Normal and Pathological Functions of Rodent and Human CNS

EAE is a rodent model that can reproduce typical MS lesions that occur due to inflammation, demyelination, and axonal damage [57]. While MS is generally characterized by loss of myelin and axons leading to progressive neurological deterioration [3,58], there are no specific tests for MS, its diagnosis often relies on ruling out other conditions that might produce similar signs and symptoms, known as a differential diagnosis. These include blood tests, magnetic resonance imaging (MRI), lumber puncture, and evoked potential tests such as visual stimuli, or electrical stimuli [59].

Numerous studies, aiming for a diagnosis of MS by examining the miRNA expressions, have detected specific miRNA ‘signatures’ for MS. For example, in peripheral blood mononuclear cells, a distinct profile of miRNA has been identified between relapse and remission states in MS and EAE, a rodent model for MS [57,60,61]. Accumulating evidence shows that the miRNAs in the CNS may play crucial roles in abnormal and normal functions in CNS. For example, overexpression of miR-134 is associated with epilepsy in experimental rodent models and temporal lobe epilepsy in humans [60]. Inhibition of miR-134 expression in the brain using antisense oligonucleotides can attenuate status epilepticus in rodents [58]. Many miRNAs are involved in immune system development and also regulate the immune system; for example, miR-150 is critical for B cell differentiation [61], miR-155 promotes autoimmune inflammation [62], and miR-146a regulates the T-cell mediated T helper-1 (Th-1) response [63] and is expressed in interleukin-17 (IL-17)-producing Th-17 cells in rheumatoid arthritis [64]. IL-17-producing Th-17 cells plays a key role in MS and other autoimmune diseases [64,65,66,67] and a recent study showed a correlation between miR-326 expression and the disease severity in MS patients as well as in EAE rats [68]. All continuing efforts to search for advanced detection technique(s) in identifying the early onset of MS may lead to a prompt and efficient therapy.

## 3. MS Therapy with Conventional, Complementary, and Alternative Medicines and the Perspective of miRNA Targeting

Treatment for MS typically focuses on gaining some recovery after attacks, slowing the progression of the disease, and managing MS symptoms. Conventional therapy varies from drugs to diet. Corticosteroids and plasmapheresis, used as a common treatment for MS, have been found through clinical trials to reduce the number of relapses and inflammation, and limit new disease with a short span of activity. Ocrelizumab (infusion administration) has been administered to reduce the number of relapses, delay progression of disability (in some patients), and limit new disease activity (seen on MRI). Other medications such as beta interferons, fingolimod, dimethyl fumerate and other drugs are used to prevent or slow down the MS progression but they have minimal effects in delaying disability progression [15]. Symptomatic medications such as muscle relaxants, fatigue medication, and antidepressants are also used to treat the specific symptoms. To maintain body fitness, typical alternative treatment strategies such as Tai Chi, chiropractor therapy, meditations, and yoga are also recommended for relieving stress [15]. A list of the therapeutic drugs and other alternative therapies for MS patients is provided in Table 1.

Although some miRNAs have been shown to promote inflammation and autoimmunity via Th cells, their precise roles in regulation of the inflammatory diseases have not yet been clearly defined. Recently, they have been used as a new therapeutic approach in the treatment and prevention of autoimmune disorders. Alteration in specific miRNA expression may produce desirable therapeutic outcomes in MS and other autoimmune diseases (Table 2). A therapeutic strategy targeting miRNA in MS and EAE relies on manipulation of endogenous miRNA levels by delivering the oligonucleotides mimicking or inhibiting the specific mRNA sequences [67]. MS and EAE are associated with overexpression of miR-155, knockdown of which results in low Th1 and Th17 cells and mild EAE [67,68]. MiR-155-3p up-regulates the Th17 population by inhibiting two heat shock protein 40 genes and this contributes to the development of EAE [69]. As miRNAs target multiple regulatory gene expressions, they may have an advantage over the concept of targeting multiple genes individually. At the same time, potential target of multiple genes is also a concern. However, these side effects could be modified and addressed by utilizing the effective methods of delivery. The miRNA therapeutic approach technology has recently been examined in different cancer models with minimum or no toxicity [70,71]. Obliteration of miR-338 aggravates the miR-219 mutant hypomyelination phenotype but miR-219 mimics enhance myelin restoration in EAE animals, indicating a therapeutic role for miR-219 in myelin repair [72].

The miRNA-mediated regulation of sphingolipid metabolism is of current scientific interest. Down-regulation of miR-101 leading to Cer synthase elevation and upregulation of Cer synthesis has been reported in metastasis-prone lung cancer cells [74]. A change in the sphingolipid compositions, along with a sub-acute change in plasma miRNA, in traumatic brain injury has also been reported [75]. A remarkable conservation of the majority of miRNA and the total lipid content indicates that they can potentially be used as biomarkers for diagnosis of diseases [76]. Despite all the recent advanced knowledge about immunopathogenesis and demyelination in MS, there is no efficacious therapy available for this devastating disease. Because our study indicates an elevation of ceramide via SPT activation leading to ceramide/sphingosine accumulation in MS [2], we propose that a therapy to block the SPT1 gene using miRNA technology will prevent the ceramide generation and this may be considered as an alternate therapeutic approach. Fingolimod (FTY720) is phosphorylated in vivo to form fingolimod-phosphate. Fingolimod-phosphate initially activates lymphocyte S1P1 via high-affinity receptor binding, yet subsequently induces S1P1 down regulation so as to prevent lymphocyte egress from lymphoid tissues, thereby reducing auto-aggressive lymphocyte infiltration into the CNS and further preventing demyelination in MS [73]. Circulating miR-15b, miR-23a and miR-223 levels were significantly down regulated in serum samples from relapsing remitting MS patients, but levels of these circulating miRNAs recovered in the MS patients following fingolimod treatment for 6 months, indicating that pharmacological manipulation of levels of miRNAs could be an emerging therapeutic strategy in reducing the frequency of exacerbations in MS patients [77]. However, our study indicates that fingolimod has much less potential in blocking Cer elevation compared to myriocin [2]. These results are important as they indicate that targeting Cer biosynthetic pathways for alteration in levels of specific miRNAs may provide significant therapeutic benefits in relapsing remitting MS patients. A detailed overview of the experimental design and findings of studies investigating miRNAs as potential biomarkers of MS have recently been published [78], which may help in designing efficient miRNAs in future MS therapy. An approach of MS therapy can be initiated by targeting SPT regulation by selecting one or more from a group of miRNAs such as miR-376, miR-30, miR-128, miR-126, miR-7, and miR-9, as these miRNAs are involved in interfering with Cer synthesis [49].

## 4. Conclusions

In summary, we have explored the dual role of Cer and Sph in mammalian brain development and diseases, specifically in MS. It is worth mentioning that Cer is a major component of sphingolipid (g/g) while Sph is a minor component (ng/g), and they play a significant role in myelinogenesis. Their roles in apoptosis in MS leading to demyelination has recently been published [2]. Sph is the base structure of a sphingolipid and displays toxicity with a minor change in concentration. Measuring the Sph/dhSph ratio in the blood of a MS patient may be utilized to predict the onset of MS, but this needs a careful and thorough investigation.

MS pathology and demyelination are still an open field for research exploring the precise cause of MS. The diagnosis of MS includes multiple procedures such as MRI, spinal fluid examination (for abnormal immune response), vision and memory assessments, nerve conduction, etc., all of which confirm the onset and progression of MS. There is a great demand for evaluating an early assessment and diagnosis of MS in order to initiate therapy at a preliminary stage of the disease. It is noteworthy that there is no single therapeutic approach that can help control and/or improve the patient’s health and, of course, there is no comprehension for cure. Most therapies are based on the patient’s symptoms and include a wide variety of medications and conventional therapies. One such difficult hurdle is our lack of knowledge regarding the precise biochemical mechanisms, the complex interactions and regulation, of oligodendrocytes, neurons, and astrocytes during myelin formation, and the participation of biomolecules such as lipids and proteins in protecting the myelin sheath. It is now certain that there are dramatic changes in the sphingolipid profile in MS brains and such changes have also been reflected in NAWM as a clear indication of the disease progression. These changes are mediated by primary biochemical alterations and/or a series of cytokines that are auto-released, either offering immunoprotection against some unidentified invasive agents that are yet to be characterized. Although current therapeutic strategies are unable to offer an absolute cure, they can help patients towards a better life by protecting the oligodendrocytes/neurons from lipid-mediated cell death. One such option includes treatment with synthetic enzyme inhibitors, but most, if not all of them, may be toxic to our system and may exert serious side effects. Alternatively, we can use miRNAs to inhibit enzyme-mRNA expression with lesser side effects using innovative technology for drug delivery. With other conventional therapeutic methods, one must develop sphingolipid miRNA technology not only for MS but also for other diseases (such as cancers) where bioactive lipids also play a devastating role.

## Figures and Tables

**Figure 1 ijms-20-05031-f001:**
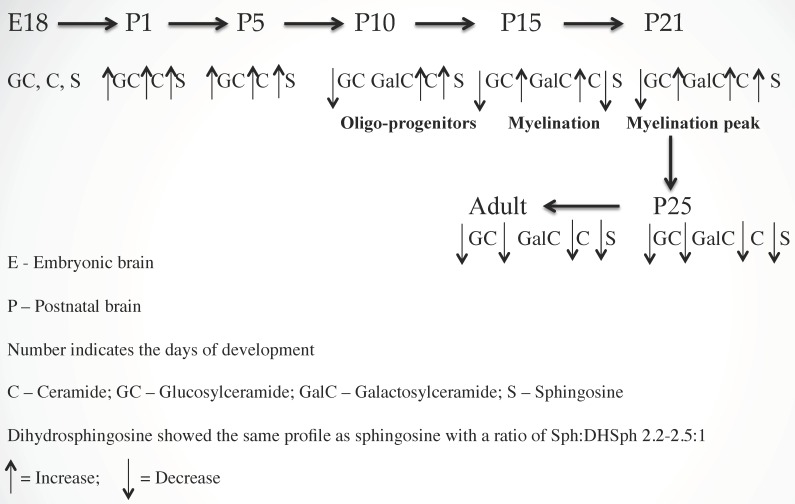
Sphingolipid regulated vertebrate brain development and myelinogenesis.

**Figure 2 ijms-20-05031-f002:**
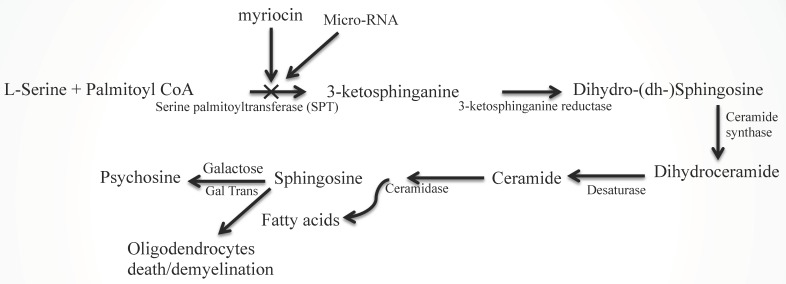
Ceramide and sphingosine generation via SPT activation and its inhibition by myriocin and miRNA. The biosynthesis of sphingolipids (dihydrosphingosine, ceramides, and sphingosine) in the de novo pathway begins with the condensation of palmitoyl-CoA and L-serine by serine palmitoyltransferase (SPT), producing 3-ketosphinganine or 3-ketodihydrosphingosine. The rate-limiting enzyme of the sphingolipid biosynthesis pathway is SPT, which can be pharmacologically inhibited by myriocin. Then, 3-ketosphinganine reductase (3-ketodihydrosphingosine reductase) reduces 3-ketosphinganine to sphinganine (dihydrosphingosine). Ceramide synthase or fatty acyltransferase converts sphinganine (dihydrosphingosine) to dihydroceramide. Dihydroceramide is dehydrogenated (desaturated) at C3–C4 of the sphinganine of dihydroceramide by desaturase to generate ceramide. The action of ceramidase converts ceramide into sphingosine via releasing fatty acids. The enzyme galactosyltransferase catalyzes the reaction between sphingosine and UDP-galactose to produce psychosine (galactosylsphingosine).

**Table 1 ijms-20-05031-t001:** A short list of common medications and other therapies for MS patients*.

Medications	Administration	Purpose	Therapeutic Approach
Methylprednisolon	Intravenous	Reduce inflammation	Lowest tolerance dose
Prednisolon	Oral	Managing the relapse	Lowest tolerance dose
ACTH	Injection	Acute exacerbation	0.75 U/m^2^ twice daily for 2 weeks
Interferon beta	Injection	Modify the course Immunosuppression	30–250 g alternate day
Glatiramer acetate	Injection	Modify the course Immunomodulator	20–40 mg/day
Teriflunomide	Oral	Delay the progression	7–14 mg once daily
Fingolimod	Oral	Delay the progression	0.25 mg then 0.5 mg
Dimethyl fumerate	Oral	Delay the progression	120 mg twice a day/1 week 240 mg twice a day
Alemtuzumab	IV infusion	Inhibit immune cell to cross BBB	10 mg/mL once for 5 days
Natalizumab	Infusion	Inhibit immune cell to cross BBB	20 mg/mL for 2 weeks
Mitoxantrone	Infusion	Inhibit immune cell to cross BBB	140 mg/m^2^
Ocrelizumab	Intravenous infusion	Block CD20 + ve lymphocytes	300 mg 1st week 300 mg 2nd week 600 mg every 6 months
Hematopoietic stem cells	Transplantation surgery (chronic progressive stage)	*De novo* generation of naïve lymphocytes to reduce autoimmunity	Once

**Symptomatic drugs for MS:** Other drugs are used to treat MS related specific symptoms such as bladder problems (oxybutynin, imipramine, etc.), infection (methenamine, sulfamethoxazole, etc.), bowel dysfunction (magnesium hydroxide, bisacodyl, etc.), depression (sertraline, fluoxetine, etc.), emotional change (dextromethorphan + quinidine), fatigue, itching, pain, sexual problem, spasticity, tremors, walking difficulty, dizziness, and vertigo. *This is tabulated from the National Multiple Sclerosis Societies website. **Alternative therapies for MS**: Tai Chi, meditations, and yoga may boost the mental concentration and body flexibility of MS patients to regain mental strength and fitness without any side effects.

**Table 2 ijms-20-05031-t002:** Emerging therapeutic strategies for alteration in levels of specific miRNAs in the treatment of MS and EAE.

Autoimmune Disease	Alteration in Specific miRNA Expression	Outcomes of Targeting Specific miRNA	Reference
MS, EAE	Upregulation of miR-326 occurred in MS patients and EAE mice.	Ets-1 expression was down regulated by miR-326 in relapsing MS patients. Disease severity was reversed in EAE mice by expressing Ets1 with a mutated 3′ UTR.	[67]
MS, EAE	Overexpression of miR-155 highly correlated with disease severity in MS patients and EAE mice.	Knockdown of miR-155 resulted in low Th1 and Th17 cells and mild EAE.	[66]
EAE	Deletion of miR-338 enhances the miR-219 mutant hypomyelination phenotype in EAE.	miR-219 mimics cooperate with miR-338 for myelin repair in EAE.	[71]
MS	Circulating miR-15b, miR-23a and miR-223 levels were decreased in relapsing remitting MS patients.	Fingolimod (FTY720) treatment recovered levels of miRNAs and reduced the frequency of exacerbations in MS patients.	[73]

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
