# Peer review of "Ceramide and Sphingosine Regulation of Myelinogenesis: Targeting Serine Palmitoyltransferase Using microRNA in Multiple Sclerosis"

_ijms, 2019, doi:10.3390/ijms20205031_

Round 1
Reviewer 1 Report
Referee comment on revision 1
“Ceramide and sphingosine regulate myelinogenesis but trigger cellular toxicity in multiple sclerosis: Protecting oligodendrocytes by targeting serine palmitoyltransferase using microRNA - a novel therapeutic concept"
Title is too long. What about the title below?
“Ceramide and sphingosine regulation of myelinogenesis by microRNA - a novel therapeutic concept for MS therapy”
Abstract: Lines 14 -15 “However, they manifest toxicity to oligodendrocytes in multiple sclerosis (MS), a human autoimmune demyelinating disease with progressive neurodegeneration.”
It is not clear to me if authors refer to an in vivo quantification of ceramide and sphingosine levels in MS brain. Please clarify this point or check English of this phrase. Personally I would delete this phrase it as it is not necessary in the contest of the abstract.
More generally I found the abstract still confusing, as too many particulars are given. Abstract should be informative on rational and the general topics that will be discussed in the review. To better clarify my view, I tentatively reordered the phrases of the abstract accordingly:
Suggested rephrased ABSTRACT
13 Ceramide and sphingosine display unique profile during brain development indicating
14 their critical role in myelinogenesis. (deletion lines 14-15)
16 Employing advanced technology such as gas chromatography–mass spectrometry, high
17 performance liquid chromatography, immunocytochemistry, along with cell culture and molecular
18 biology we have previously reported sphingosine elevation in MS leading to oligodendrocyte death fostering
19 the demyelination. Ceramide elevation by serine palmitoyltransferse (SPT) activation was the prime
20 source to sphingosine as myriocin, an inhibitor of SPT, protected oligodendrocytes inhibiting
21 sphingosine elevation. Supporting this view, Fingolimod, a drug used for MS therapy, reduced ceramide generation thus
22 offering a partial protection. (Deletion 23-24)
25 Sphingolipid synthesis and degradation in normal development is regulated by a series of
26 miRNAs and hence, accumulation of sphingosine in MS may be prevented by employing miRNA
27 technology. Designing a precise therapeutic approach for MS by SPT inhibition via up- or down
28regulation of miRNAs may prevent ceramide generation and provide protection to
29 oligodendrocytes. In this review it will be discussed the current knowledge regarding how Ceramide and Sphingolipid synthesis and degradation and how they could be regulated by miRNA to be used in therapeutic approaches for MS
Introduction
Page 2 Lines 39-42 “in this article we have introduced the concept of miRNA therapy …..” They are referring to this “review” in which they will discuss the potential use of miRNA therapy in MS or they are referring to an article they might have written previously in which they discussed this point ? Please specify, in case or correct the word article with “review”
Lane 50 “Other chemical inhibitors” phrase too generic, it can be deleted or references should be given if known
Lane 1 pag 3 “this article…”= This review?
I disagree that the review describes “the plausible application….” as it describes the basic rational on which miRNA therapies based on regulation of Ceramide and Sphingosine could be applied to MS therapy, if I well understood.
Figure 1: Arrows have been copy pasted on the top of text! This is out of journal standard for figure construction and arrow make the text not understandable. Figure should be produced following journal standards
A similar problem is in Figure 2.
Minor point:Pag 7 lines 1-9 References are missing
Please make sure that references are updated
Author Response
We thank the reviewer for constructive criticisms and this manuscript has been considerably improved following the revision. The following is a point-by-point response to the reviewer’s criticisms.
Criticism1. Title is too long. What about the title below? “Ceramide and sphingosine regulation of myelinogenesis by microRNA - a novel
therapeutic concept for MS therapy”
Response: We have revised the title in short form following the suggestion. The new title is – “Ceramide and sphingosine regulation of myelingensis: Targeting serine palmityoltransferase using micro RNA in multiple sclerosis. We like to carry SPT as this enzyme is the target of miRNA therapy.
Criticism 2: Abstract: Lines 14 -15 “However, they manifest toxicity to oligodendrocytes in multiple sclerosis (MS), a human autoimmune
demyelinating disease with progressive neurodegeneration.” It is not clear to me if authors refer to an in vivo quantification of ceramide and sphingosine levels in MS brain. Please clarify this point or check English of this phrase. Personally I would delete this phrase it as it is not necessary in the contest of the abstract.
Response: We have deleted the phrase.
Criticism 3: More generally I found the abstract still confusing, as too many particulars are given. Abstract should be informative on rational and the general topics that will be discussed in the review. To better clarify my view, I tentatively reordered the phrases of the abstract accordingly: Suggested rephrased ABSTRACT have revised the abstract to clarify our objective.
Response: We have considered all suggestions and revised the abstract by inserting the proposed changes with some modifications for further clarification.
Criticism 4: Introduction - Page 2 Lines 39-42 “in this article we have introduced the concept of miRNA therapy …..” They are referring to this “review” in which they will discuss the potential use of miRNA therapy in MS or they are referring to an article they might have written previously in which they discussed this point? Please specify, in case or correct the word article with “review”
Response: We have corrected the sentences by rephrasing the paragraph for further clarity.
Criticism 5: Lane 50 “Other chemical inhibitors” phrase too generic, it can be deleted or references should be given if known
Response: We have revised the sentence and added a new reference (#19).
Criticism 6: Lane 1 page 3 “this article…”= This review?
Response: Inserted the suggested change.
Criticism 7: I disagree that the review describes “the plausible application….” as it describes the basic rational on which miRNA therapies based on regulation of Ceramide and Sphingosine could be applied to MS therapy, if I well understood.
Response: Following the criticism we have rephrased the sentence to comply with the reviewer’s comment.
Criticism 8: Figure 1: Arrows have been copy pasted on the top of text! This is out of journal standard for figure construction and arrow make the text not understandable. Figure should be produced following journal standards. A similar problem is in Figure 2.
Response: The figure 1 is now newly drawn and the figure 2 was also checked for correction.
Criticism 9: Minor point: Page 7 lines 1-9 References are missing
Response: We have added the reference.
Criticism 10: Please make sure that references are updated
Response: The references have been rechecked and revised.
Reviewer 2 Report
The revised version of the manuscript seems much improved compared with the original version. According to criticism#2, the manuscript has been appropriately edited and references have been corrected. However, I cannot find a paragraph describing the miRNA manipulation of sphingolipid metabolism in response to criticism #1. Please propose which miRNAs could be used to manipulate sphingolipid metabolism for MS therapy.
[minor comments]
#1. (page 2, line 36) a ligand for Sph-1-P “receptor” (“receptor” is missing.)
#2. Texts in Figures 1 and 2 do not appear properly.
#3. (page 9, line 23) the patient’s “syndrome” should be “symptom”.
#4. Reference #50 is the same as #10.
#5. (page 13, line 3) “RNAsas” should be “RNAs as”.
Author Response
We thank the reviewers for constructive criticisms and this manuscript has been considerably improved following the revision. The following is a point-by-point response to the reviewers’ criticisms.
Criticism 1: The revised version of the manuscript seems much improved compared with the original version. According to criticism#2, the manuscript has
been appropriately edited and references have been corrected. However, I cannot find a paragraph describing the miRNA manipulation of sphingolipid
metabolism in response to criticism #1. Please propose which miRNAs could be used to manipulate sphingolipid metabolism for MS therapy.
Response: We thank the reviewer for acknowledging the improvement and the response. The miRNA regulation of sphingolipid synthesis has been described under the title “miRNAs in oligodendrocyte development and in ceramide metabolism (2.4.1).
Criticism 2: (page 2, line 36) a ligand for Sph-1-P “receptor” (“receptor” is missing.)
Response: The word “receptor” has been inserted.
Criticism 3: #2. Texts in Figures 1 and 2 do not appear properly
Response: Both figures have revised and inserted properly.
Criticism 4: (page 9, line 23) the patient’s “syndrome” should be “symptom”.
Response: The word is corrected.
Criticism 5: Reference #50 is the same as #10.
Response: The reference #50 has been corrected and a new reference (#19) has been added in this revision.
Criticism 6: (page 13, line 3) “RNAsas” should be “RNAs as”.
Response: Verified and corrected.
Round 2
Reviewer 1 Report
the review has improved a lot from the first version. I still think that the review rises an interesting and novel issue but in my opinion the manuscript still requires a professional scientific editing to revise the style and some minor typing mistakes
Just to give you an example I report below one section of the manuscript that in my opinion would gain from such professional English editing, as the phrases nor the title, follow one another.
I am very sorry to give you further comments, but as I wrote at my first review, the manuscript although interesting was really written with no attention or accuracy for details.
I am not English mother language, so I cannot help in English scientific editing as reviewer.
Please do not consider me for further review of this article as I think that the manuscript could have been corrected in many parts by the authors prior to the first submission , to avoid that the reviewers would have to request so many corrections on English style and figures.
In any case please read below for the correction I suggest for
section 2.4.3
2.4.3. Role of miRNAs in EAE and immunity
29 EAE is a rodent model that can reproduce typical MS lesions that occur due to inflammation,
30 tdemyelination, and axonal damage [58]. While MS is generally characterized by loss of myelin, and
31 axon leading to progressive neurological deterioration [3, 59], there are no specific tests for MS, its
32 diagnosis often relies on ruling out other conditions that might produce similar signs and
33 symptoms, known as a differential diagnosis. These include blood test, magnetic resonance imaging
34 (MRI), lumber puncture, and evoked potential tests such as visual stimuli, or electrical stimuli [60].
35 The miRNAs in the CNS may play crucial roles in abnormal and normal functions in CNS. For
36 example, overexpression of miR-134 is associated with epilepsy in experimental animal models and
37 temporal lobe epilepsy in humans [61]. Inhibition of miR-134 expression in the brain using
38 antisense oligonucleotides can attenuate status epilepticus in rodents [60]. Many miRNAs are
39 involved in immune system development and also regulate the immune system; for examples,
40 miR-150 is critical for B cell differentiation [62], miR-155 promotes autoimmune inflammation [63],
41 and miR-146a regulates T-cell mediated T helper-1 (Th-1) response [64] and it is expressed in
42 interleukin-17 (IL-17) producing Th-17 cells in rheumatoid arthritis [65].
43 Numerous studies aiming for diagnosis of MS by examining the miRNA expressions have
44 detected specific miRNA ‘signatures’ for MS; for example, in peripheral blood mononuclear cells a
45 distinct profile of miRNA has been identified between relapse and remission states in MS and EAE
46 [61, 62]. Growing evidence suggests that IL-17 producing Th-17 cells plays a key role in MS and
47 other autoimmune diseases [65-68] and a recent study showed a correlation between miR-326
48 expression and the disease severity in MS patients and in EAE rats [69]. All continuing efforts to
Here is how I would have written the above section : that I would have entitled
Role of miRNA in normal and pathological function of rodent and human CNS
“MS is generally characterized by loss of myelin, and
31 axon leading to progressive neurological deterioration [3, 59], there are no specific tests for MS, its
32 diagnosis often relies on ruling out other conditions that might produce similar signs and
33 symptoms, known as a differential diagnosis. These include blood test, magnetic resonance imaging
34 (MRI), lumber puncture, and evoked potential tests such as visual stimuli, or electrical stimuli [60].
43 Numerous studies aiming for diagnosis of MS by examining the miRNA expressions have
44 detected specific miRNA ‘signatures’ for MS; for example, in peripheral blood mononuclear cells a
45 distinct profile of miRNA has been identified between relapse and remission states in MS and EAE (
46 a rodent model for MS [58] [61, 62].
35 Accumulating evidences show that The miRNAs in the CNS may play crucial roles in abnormal and normal functions in CNS. For
36 example, overexpression of miR-134 is associated with epilepsy in experimental animal models (WHICH?) and
37 temporal lobe epilepsy in humans [61]. Inhibition of miR-134 expression in the brain using
38 antisense oligonucleotides can attenuate status epilepticus in rodents [60]. Many miRNAs are
39 involved in immune system development and also regulate the immune system; for examples,
40 miR-150 is critical for B cell differentiation [62], miR-155 promotes autoimmune inflammation [63],
41 and miR-146a regulates T-cell mediated T helper-1 (Th-1) response [64] and it is expressed in
42 interleukin-
Author Response
We thank the reviewer for constructive criticisms and this manuscript has been considerably improved following the revision. The changes proposed by the reviewer have been adopted and highlighted in the text. Please read precisely.
Thank you so much.
This manuscript is a resubmission of an earlier submission. The following is a list of the peer review reports and author responses from that submission.
Round 1
Reviewer 1 Report
In this review article, the authors summarize the involvement of sphingolipids and microRNAs in multiple sclerosis, and propose the application of microRNAs targeting sphingolipid metabolism as a novel therapeutic approach for multiple sclerosis. Although the concept seems intriguing, the plausible methods of manipulating sphingolipid metabolism by specific microRNAs are not presented in the manuscript. In addition, English editing may be needed throughout the manuscript (for example, lines 1-6 in page 6 seem to be misplaced, and not a few references are inappropriately numbered). Therefore, major revision should be required for publication in International Journal of Molecular Sciences.
Reviewer 2 Report
The authors intend to write a review on the role of Sphingolipid accumulation in MS disease development and to put forward the idea that by targeting this pathway, current therapeutical intervention could be improved.
However, it is hard from the abstract to understand the rational and topic that they will discuss in the review, as essentially the abstract describes some of their findings.
Introduction : A scheme of the sphigolipid role in brain development and myelination could help the reader if placed at the beginning of the introduction. This could be a scheme showing how sphigosine toxicity could mediated the effects in MS brain described in literature .
In general the introduction is not clear and topics do not follow one to the other with a clear rational. A description of the rational on which the topics will be folloed in the chapters of the review would help.
There is a paragraph .2 entitled DISCUSSION. This title is not appropriate for a review.
In general, the major problem of the manuscript is that instead to review current understanding on the chosen topic, it is full of “we propose” we “anticipate” phrases that are not sustained by the data or references that they discuss. Nor I think a review should be based on personal opinions, expressed in such a way. It should review the current knowledge and leave the reader derive its own conclusion.
There are fonts changes in the text and many typos errors